# Evaluation of a multiplex PCR method for the detection of porcine parvovirus types 1 through 7 using various field samples

Seung-Chai Kim[1][☯], Chang-Gi Jeong[1][☯], Salik Nazki[1,2], Sim-In Lee[1,3], Ye-Chan Baek[1], Yong-Jin Jung[1], Won-Il Kim[1]*

**1** College of Veterinary Medicine, Jeonbuk National University, Iksan, Korea, **2** The Pirbright Institute, Pirbright, United Kingdom, **3** Animal and Plant Quarantine Agency, Gimcheon, Korea

☯ These authors contributed equally to this work.

* kwi0621@jbnu.ac.kr

**Data Availability Statement:** All relevant data are within the paper and its Supporting Information files.

## Abstract

Porcine parvoviruses (PPVs) are small, nonenveloped DNA viruses that are widespread in the global pig population. PPV type 1 (PPV1) is a major causative agent of reproductive failure and has been recognized since the 1960s. In recent decades, novel PPVs have been identified and designated as PPVs 2 through 7 (PPV2~PPV7). Although the epidemiological impacts of these newly recognized parvoviruses on pigs are largely unknown, continuous surveillance of these PPVs is needed. The aim of this study was to develop an improved and efficient detection tool for these PPVs and to assess the developed method with field samples. Using 7 sets of newly designed primers, a multiplex polymerase chain reaction (mPCR) protocol was developed for the simultaneous detection of the seven genotypes of PPV (PPV1~PPV7). The sensitivity of the mPCR assay was analyzed, and the detection limit was determined to be $3 \times 10^3$ viral copies. The assay was highly specific in detecting one or more of the viruses in various combinations in specimens. The mPCR method was evaluated with 80 serum samples, 40 lung or lymph node samples and 40 intestine or fecal samples. When applied to these samples, the mPCR method could detect the 7 viruses simultaneously, providing rapid results regarding infection and coinfection status. In conclusion, the developed mPCR assay can be utilized as an effective and accurate diagnostic tool for rapid differential detection and epidemiological surveillance of various PPVs in numerous types of field samples.

## Introduction

Parvoviruses are members of the *Parvoviridae* family, which has an extensive host range and can be divided into two subfamilies: *Parvovirinae* and *Densovirinae*. The members of *Parvovirinae* infect vertebrate hosts, while those of *Densovirinae* infect arthropods. *Parvovirinae* comprises eight genera: *Protoparvovirus*, *Tetraparvovirus*, *Copiparvovirus*, *Chapparvovirus*, *Dependoparvovirus*, *Erythroparvovirus*, *Bocaparvovirus* and *Amdoparvovirus* [1, 2]. The first four genera contain porcine parvoviruses (PPVs). The present taxonomy proposed by the

**Funding:** This work was supported by the Korea Institute of Planning and Evaluation for Technology in Food, Agriculture, Forestry (IPET) through the Animal Disease Management Technology Development Program funded by the Ministry of Agriculture, Food and Rural Affairs (MAFRA) (118093-03).

**Competing interests:** The authors have declared that no competing interests exist.

International Committee on Taxonomy of Viruses (ICTV) classifies PPV1 into *Protoparvovirus*; PPV2 and PPV3 into *Tetraparvovirus*; PPV4, PPV5 and PPV6 into *Copiparvovirus*; and PPV7 into *Chapparvovirus* [3, 4].

PPVs are packaged in a nonenveloped icosahedral capsid with single-stranded linear DNA approximately 4 to 6.3 kb in length [1]. The genomes of PPVs consist of two gene cassettes, each of which contains the P4 and P40 promoters [2, 5]. The nonstructural protein(s) (NS[s]) are transcribed from the left open reading frame (ORF) by the P4 promoter, and these proteins have replicase activity. The right ORF encodes the structural protein(s) (VP[s]), which are transcribed from the P40 promoter and are composed of capsids [2, 5, 6]. An additional ORF, ORF3, translates nuclear phosphoproteins (NPs) and is located in the middle of ORF1 and ORF2; this ORF is characteristic of members of the *Bocaparvovirus* genus and of PPV4 [6, 7]. PPVs are considered to have more stable genomes than other parvoviruses and ssDNA viruses [5, 8–10]. High mutation rates of approximately $3 \sim 5 \times 10^{-4}$ have been observed in the VP genes, while moderate evolution rates of approximately $10^{-5}$ have been observed in NS genes [5].

PPV1 was the sole representative of *Parvovirinae* members until recently and has been ubiquitous in the global pig population as a major causative agent of reproductive failure in pigs. Recently, Several PPVs have been newly identified by molecular methods and have been sequentially designated PPV2 through PPV7, but the true impact of these novel parvoviruses on swine health has not been defined clearly given that virus isolation and experimental infections have not been performed [3, 4, 6, 11–14]. Multiplex PCR (mPCR) methods have been developed for the simultaneous detection of various swine pathogens, including PPVs [15]. Furthermore, PPV1 through PPV6 have been detected using single, duplex or triplex differential real-time PCR assays [4, 11, 16]. PPV7 has also been detected by employing conventional PCR and real-time PCR methods [3, 17]. However, there is no useful and specific diagnostic assay capable of concurrently differentiating among the 7 PPVs (PPV1 through PPV7). Therefore, in the present study, a simple, specific and sensitive mPCR assay was developed to detect and differentiate among these PPVs in various samples.

## Materials and methods

### Cells, viruses and bacterial strains

Porcine reproductive and respiratory syndrome virus (PRRSV)-1 isolate CBNU0495 (KY434183.1) was grown in primary cultures of porcine alveolar macrophages (PAMs), while PRRSV-2 strain VR2332 (AY150564.1) was propagated in MARC-145 cells (African green monkey cells) at a multiplicity of infection (MOI) of 0.001. Both PAMs and MARC-145 cells were used for virus culture in RPMI 1640 medium supplemented with 10% heat-inactivated fetal bovine serum (FBS; Invitrogen), 2 mM L-glutamine, and 100× antibiotic-antimycotic solution (Anti-anti, Invitrogen; a 1× solution contains 100 IU/ml penicillin and 100 μg/ml Fungizone$^{®}$ [amphotericin B]) at 37°C in a humidified 5% $CO_2$ atmosphere. A porcine circovirus type 2 (PCV2) isolate was grown in PK15 cells (ATCC-CCL31). The PK15 cells were grown in MEM-α supplemented with 10% heat-inactivated FBS, 2 mM L-glutamine (Invitrogen) and 100× Anti-anti. Japanese encephalitis virus (JEV, strain Beijing-1; kindly provided by Dr. SK Eo, Jeonbuk National University, Iksan) was propagated in BHK-21 cells at an MOI of 1 with 2% DMEM. Pseudorabies Virus (PRV, strain Yangsan; kindly provided by DR. SK Eo) was propagated in PK15 cells with same growth medium condition used for PCV2 propagation. The PED/TGE/Rota vaccine (Choong Ang Vaccine Laboratories Co., Ltd.) and the CSF vaccine (KOMIPHARM INTERNATIONAL Co. Ltd.) were used for isolation of nucleic acids. *E. coli* field isolate maintained within the laboratory was cultured in Luria-Bertani (LB) agar

(Difco Laboratories, Inc.) and LB broth (Difco Laboratories, Inc.), whereas the *Salmonella enterica* isolate was grown in tryptic soy agar (TSA) and TS broth (Difco Laboratories, Inc.).

## Samples and nucleic acid preparation

A total of 160 samples were used in this study. All samples were submitted to the Jeonbuk National University Veterinary Diagnostic Center (JBNU-VDC) from Korean field swine farms from 2018~2019. The samples included 80 serum samples collected from 22 farms, 40 lung or lymph node samples from 28 farms, and 40 intestine or fecal samples from 17 farms.

Viral nucleic acids were extracted from 200 μl of each serum samples using an NP968 Nucleic Acid Extraction System (XI'AN TIANLONG Science & Technology Co.) according to the manufacturer's instructions. One gram of tissue samples, including lung, lymph node, intestine and fecal samples, was homogenized with mechanical homogenizer (TissueRuptor; Qiagen), mixed with 10 ml of phosphate-buffered saline (PBS; 0.1 M, pH 7.4) and centrifuged at 2,500 rpm for 10 min at 4˚C. Viral nucleic acid was immediately extracted from the supernatant using a Patho Gene-spin DNA/RNA Extraction Kit (iNtRON Biotechnology, Inc.) according to the manufacturer's instructions. All extracts were stored at -80˚C until use.

## Primer design

To design primers for conserved regions in different types of PPVs, the complete sequences of PPV1 (n = 16), PPV2 (n = 18), PPV3 (n = 14), PPV4 (n = 13), PPV5 (n = 17), PPV6 (n = 29), and PPV7 (n = 30) were collected from the NCBI GenBank (S1 Table) using CLC Sequence Viewer 8 software (https://www.qiagen.com/) and compared by multiple alignment using Lasergene® MegAlign software (DNASTAR, Inc.). The primers were designed manually by selecting the consensus region within each type of PPV. The BLAST tool from NCBI (https://blast.ncbi.nlm.nih.gov/Blast.cgi) and Multiple Primer Analyzer provided by Thermo Fisher Scientific [18] were used to validate the suitability of the designed primers. The primers (Table 1) were synthesized by BioD Co., Ltd. (Gyeonggi-do, Korea).

## Construction of plasmids for positive controls

First, 40 serum samples and 20 lung or lymph node samples were selected for simplex PCR amplification with the newly designed primers. The reaction for each PPV was performed in a

**Table 1. Primers designed for the mPCR method used to detect seven PPV types in this study.**

| Primer | Nucleotide sequence (5' - 3') | Product size (bp) | Target | Reference strain | Position |
|---|---|---|---|---|---|
| **PPV1-mF** | AGTTAGAATAGGATGCGAGGAA | 163 | NS1 | NC_001718 | 1761–1782 |
| **PPV1-mR** | TGCTTGGTAACCTTTCTTTACC | | | | 1923–1902 |
| **PPV2-mF** | GCGTGCTCAAGCTGTACC | 286 | NS1 | NC_025965 | 195–212 |
| **PPV2-mR** | CTCACTGCGAGATGAAGG | | | | 480–463 |
| **PPV3-mF** | GCTGATAGGTTGATGAATAAGGAG | 498 | VP | KY586143 | 2990–3013 |
| **PPV3-mR** | CCGCATACCCATAACAGG | | | | 3487–3470 |
| **PPV4-mF** | CTGAGACTGAATTCATCCCTG | 592 | ORF3 | GU938965 | 2347–2367 |
| **PPV4-mR** | ATCAGAATCATGTATGGTCTGC | | | | 2938–2917 |
| **PPV5-mF** | AACCGAGTTAAGAACTTTACCG | 945 | VP | JX896322 | 2773–2793 |
| **PPV5-mR** | ACCCAAGTCAGGAGTTCG | | | | 3941–3924 |
| **PPV6-mF** | GTGATAATGATGTGACTACGGAG | 396 | NS1 | MG345036 | 958–980 |
| **PPV6-mR** | CAGCAGTATGTGCAATAGCA | | | | 1353–1334 |
| **PPV7-mF** | AGGAAATGGAACATCCAGG | 802 | NS1 | NC_040562 | 505–523 |
| **PPV7-mR** | TTATCTTTCGTGGCGTCC | | | | 1306–1289 |

25 μl volume composed of 12.5 μl of 2× F-Star Taq PCR Master Mix (BIOFACT Co.), 2 μl of template, 1 μl each of the forward and reverse primers (20 μM) and 8.5 μl of nuclease-free water (NFW). The simplex PCR conditions were as follows: predenaturation at 94˚C for 1 min; 35 cycles of denaturation at 94˚C for 30 s, annealing at 60˚C for 30 s, and extension at 72˚C for 1 min; and a final extension at 72˚C for 5 min. The PCR products were evaluated with 1% agarose gel electrophoresis.

All PPV types were successfully amplified from the samples without any nonspecific bands by simplex PCR, and the positive samples for each PPV type were amplified again and purified with a Wizard® SV Gel and PCR Clean-Up Kit (Promega Co.) according to the manufacturer's instructions. Each amplicon was ligated into a pGEM®-T Easy vector (Promega Co.) using an RBC Rapid Ligation Kit (Real Biotech Co.) and transformed into DH5α HIT Competent Cells (Real Biotech Co.). Monoclonal bacterial strains for each TA-cloned PPV type were cultured, and the extracted plasmids (extracted using an Exprep™ Plasmid SV Mini Kit, GeneAll Biotechnology Co.) were sequenced using a commercial sequencing service (Macrogen Inc., Seoul, Korea). The obtained sequences were assembled using Seqman™ (DNASTAR Inc.), and the consensus sequences were verified to belong to each PPV type with the NCBI BLAST tool. The acquired consensus sequences were submitted to NCBI GenBank under the accession number MW401540-MW401546. TA clones of each PPV type (pGEM-PPV1, pGEM-PPV2, pGEM-PPV3, pGEM-PPV4, pGEM-PPV5, pGEM-PPV6, and pGEM-PPV7) were used as standard plasmids for subsequent establishment of the mPCR method.

## Optimization of the mPCR method

All standard plasmids were mixed, and the mixture was used as a template to optimize the annealing temperature (Ta). The primers were mixed into a premix of forward and reverse primers with a concentration of 10 μM for each primer. The reaction for detecting of all PPV types was performed with AccuPower® Multiplex PCR PreMix (Bioneer Co.), 1 μl of each forward/reverse primer premix, 50 ng of mixed template (pGEM-PPV1~pGEM-PPV7), and NFW. Gradient mPCR was performed with gradient Tas ranging from 50˚C to 70˚C for each PCR mixture. The mPCR conditions were as follows: predenaturation at 95˚C for 10 min; 30 cycles of denaturation at 95˚C for 30 s, varied Ta for 40 s, and extension at 72˚C for 1 min; and a final extension at 72˚C for 5 min. The PCR products were evaluated with 2% agarose gel electrophoresis.

## Specificity, sensitivity, and reproducibility of the mPCR method

To verify the specificity of the mPCR method, RNA or DNA extracted from PK-15 cells or from other pathogens (PRRSV, PCV2, porcine epidemic diarrhea virus [PEDV], transmissible gastroenteritis virus [TGEV], rotavirus, JEV, PRV, CSFV, *E. coli* and *Salmonella enterica*) was used as a template to detect possible cross-reactivity of the primers. The nucleic acid extracts from PK-15 cells, PCV2, PRV, *E. coli* and *Salmonella enterica* were used directly as templates for the specificity assay, while the products of viral RNA extraction from PRRS, PEDV, TGEV, rotavirus, JEV, and CSFV were reverse-transcribed into complementary DNA (cDNA) using a high-capacity cDNA reverse transcription kit (Applied Biosystems) and utilized as PCR templates. Both premixed plasmids and the individual plasmids were tested.

To evaluate the sensitivity of the assay, a NanoDrop instrument was utilized to measure the concentrations of the seven plasmids (pGEM-PPV1~pGEM-PPV7). The copy number of the plasmid DNA was calculated as previously described by other study groups [19, 20].

Each individual plasmid of the seven plasmids was serially diluted from $3\times10^9$ to $3\times10^1$ copies/μl, and the plasmid mixture was diluted from $3\times10^8$ to $3\times10^1$ copies/μl. The prepared

diluents were used as the templates to identify the minimum detection limit of the new PCR method.

The reproducibility of the established assay was also monitored. Using templates of plasmid mixtures diluted from $3\times10^5$ to $3\times10^3$ copies/µl, the established mPCR method was assayed three times by using three different PCR machines at different times.

## Evaluation of field samples and comparison of the mPCR method with other detection PCR methods

After establishment of the mPCR assay, all samples described above (80 serum samples, 40 lung or lymph node samples and 40 intestine or fecal samples) were tested to evaluate the capability of the method to detect PPV1~PPV7 from field samples of various types.

To validate the established assay in this study, 4 samples from among samples that were PCR-positive for each PPV type were selected. A total of 28 samples that were positive for single or multiple PPV types were tested by simplex PCR with previously published primer sets (S2 Table) [21–27].

## Results

### Establishment of the mPCR method

The optimal reaction conditions for the mPCR method were established by adjusting the primer concentration, cycle number and Ta. Premixed standard plasmids of each PPV type were used as templates, and the products were verified with 2% agarose gel electrophoresis. The optimal Ta was evaluated with the gradient option, and there were no primer dimers or nonspecific bands at 60˚C, which was selected as the optimum Ta value (Fig 1).

### Specificity of the mPCR method

The specificity of the established mPCR method was assayed using templates from sources including PK-15 cells and other normal swine pathogens (PRRSV, PCV2, PEDV, TGEV, rotavirus, JEV, PRV, CSFV, *E. coli* and *Salmonella enterica*). The new mPCR assay was able to

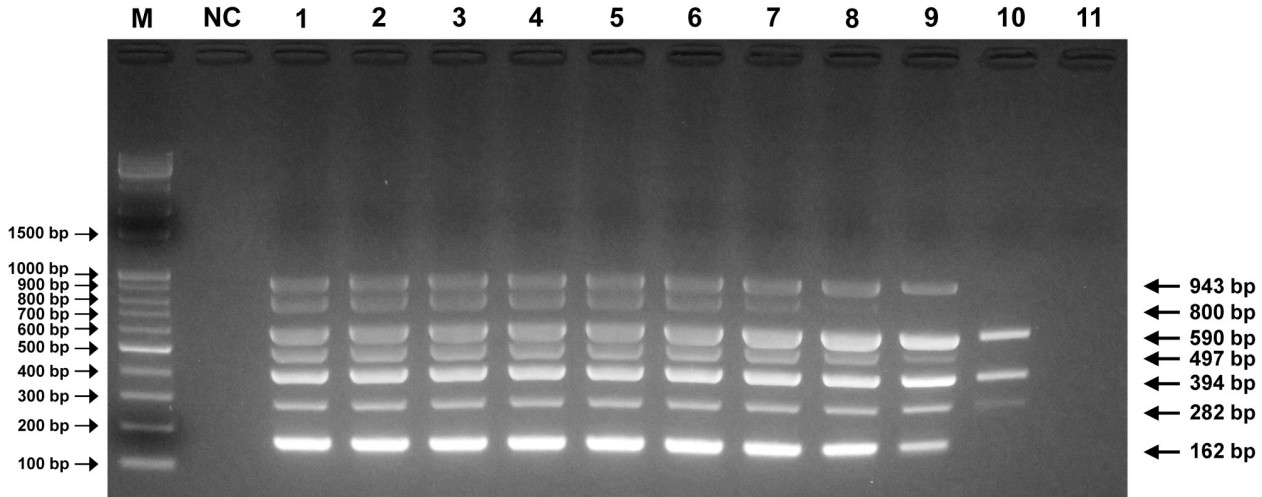

**Fig 1. Optimal Ta determination for PPV mPCR.** Agarose gel electrophoresis (2%) of standard positive controls (pGEM-PPVs). Lane M, 100 bp-plus DNA ladder; lane NC, negative control for detection of PPV1~PPV7; lanes 1~11, gradient Tas of 50.0˚C, 51.9˚C, 53.8˚C, 56.1˚C, 58.0˚C, 60.0˚C, 62.0˚C, 63.8˚C, 66.1˚C, 68.0˚C and 70˚C, respectively.

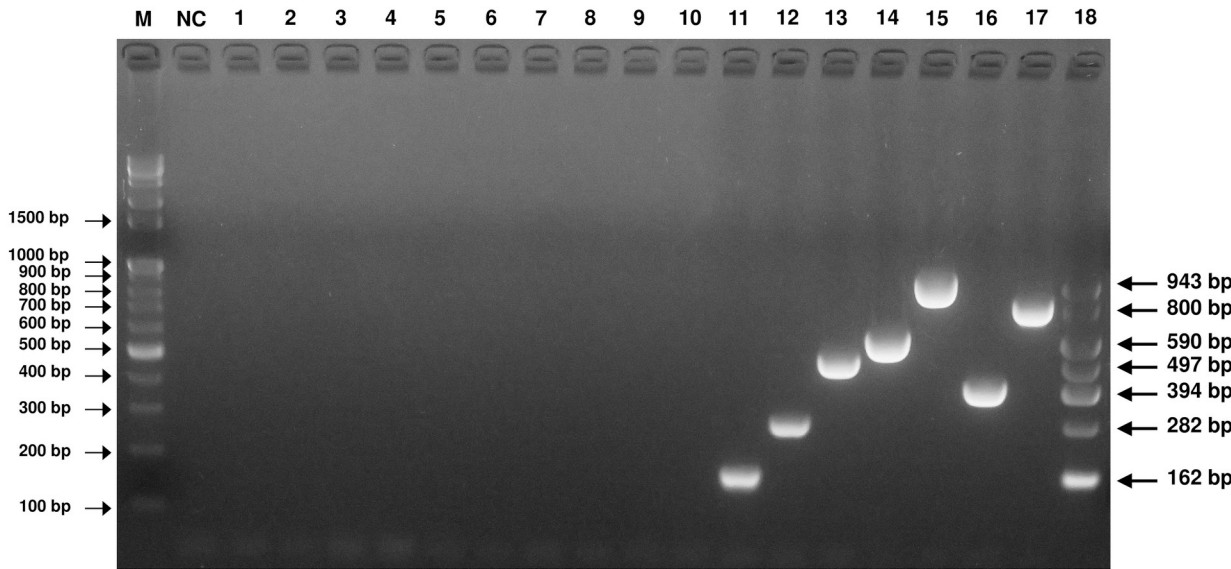

**Fig 2. Specificity of the PPV mPCR method.** Agarose gel electrophoresis (2%) of specific fragments amplified by mPCR from the proviral DNA and cDNA of Pk-15 cells, PRRS (NA), PRRS (EU), PCV2, PED/TGE/Rota-mixed (vaccine), *E. coli*, *Salmonella enterica* and JEV. Lane M, 100 bp-plus DNA ladder; lane NC, negative control; lane 1, Pk-15 cells; lane 2, PRRS (NA); lane 3, PRRS (EU); lane 4, PCV2; lane 5, PED/TGE/Rota-mixed (vaccine); lane 6, *E. coli*; lane 7, *Salmonella enterica*; lane 8, JEV; lane 9, PRV; lane 10, CSFV; lanes 11~17, pGEM-PPV1, pGEM-PPV2, pGEM-PPV3, pGEM-PPV4, pGEM-PPV5, pGEM-PPV6 and pGEM-PPV7; lane 18, mixed standard of all pGEM-PPV plasmids.

detect and distinguish among PPV1~PPV7. The amplicons of the PPVs were confirmed by sequencing (Fig 2). In contrast, no specific amplicons were produced from other templates from sources including PK-15 cells and other normal swine pathogens.

## Sensitivity of the mPCR method

The sensitivity of the mPCR method was evaluated by using each standard plasmid. The simultaneous minimum detection threshold for mPCR was $3\times10^3$ viral DNA copies, and $3\times10^2$ viral copies were detectable for several PPVs. The minimum detection limit for single PCR was $3\times10^3$ viral copies for PPV1 through PPV7. In the cases of PPV4, PPV5 and PPV6, $3\times10^2$ viral copies were detectable by single PCR (Fig 3).

## Reproducibility of the mPCR method

To verify the reproducibility of the mPCR assay, the nucleic acids of the mixed PPVs were diluted to $3\times10^3$ to $3\times10^5$ copies/μl for amplification using three different PCR machines at different times. The mixed nucleic acids of PPVs produced distinct amplicons under different conditions and produced similar results (S1 Fig).

## Cross-validation with other PCR detection methods

Twenty-eight clinical samples determined to be positive by mPCR were assessed with simplex PCR assays with published primer sets. The results were compared between the simplex PCR and mPCR methods, and the findings were as follows (number of positive diagnoses from mPCR/number of positive diagnoses from simplex PCR): for PPV1, 5/5; for PPV2, 11/12; for PPV3, 8/9; for PPV4, 5/5; for PPV5, 8/8; for PPV6, 11/10; and for PPV7, 5/4. Thus, the results of the mPCR assay and simplex PCR assay showed 100% agreement in the total number of

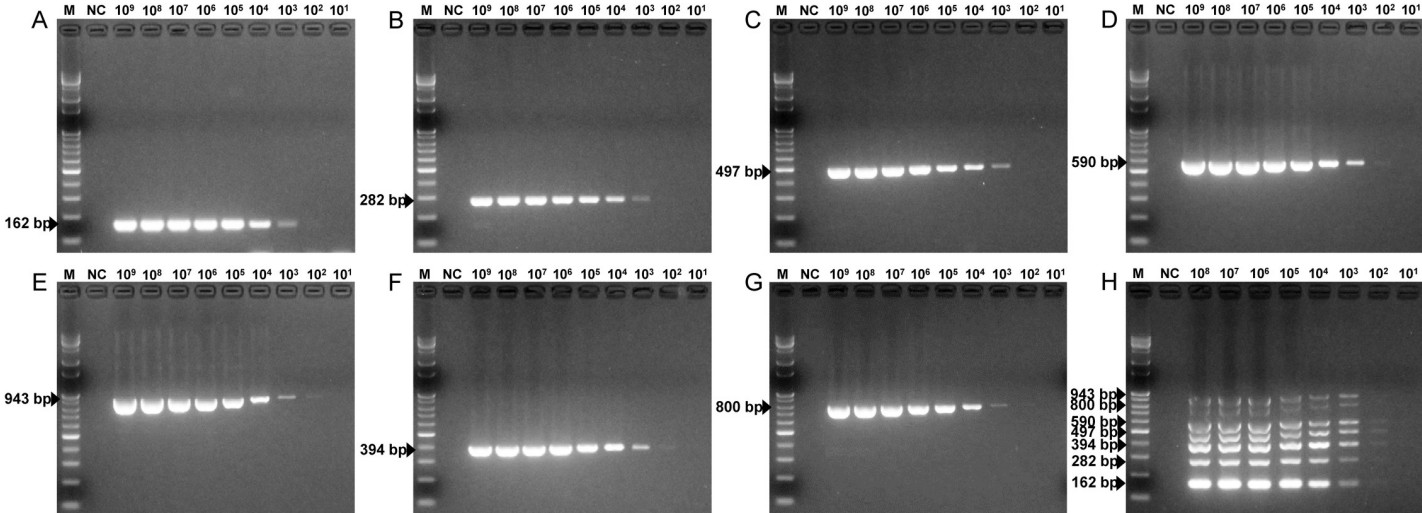

**Fig 3. Sensitivity of the PPV mPCR method.** The seven pGEM-PPV single plasmids, diluted from $3\times10^9$ to $3\times10^1$ copies/µl, and the mixed plasmids, diluted from $3\times10^8$ to $3\times10^1$ copies/µl, were used to determine the minimum detection limit of the PPV mPCR method. (A) Sensitivity for pGEM-PPV1. (B) Sensitivity for pGEM-PPV2. (C) Sensitivity for pGEM-PPV3. (D) Sensitivity for pGEM-PPV4. (E) Sensitivity for pGEM-PPV5. (F) Sensitivity for pGEM-PPV6. (G) Sensitivity for pGEM-PPV7. (H) Sensitivity for pGEM-PPV1~pGEM-PPV7. Lane M, 100 bp-plus DNA ladder; lane NC, negative control.

tests. However, mPCR was more efficient with regard to time, effort and reagent cost for the detection of PPVs than simplex PCR.

## Evaluation of field samples

The ability of the mPCR method to detect PPVs in different sample types was tested with 80 serum samples, 40 lung/lymph node samples and 40 intestine/fecal samples. The mPCR performed well for all types of samples and produced distinct amplicons for the clinical samples (S2 Fig). The results of mPCR are summarized in Table 2.

Among serum samples, 53.3% (48/80) of the samples were positive for PPVs. The rate of single infection (35.0%, 28/80) was higher than that of multiple infection (25.0%, 20/80) in serum. The single infection rate was highest for PPV6 (12.5%, 10/80) followed by PPV2 (11.3%, 9/80). Single infection with PPV3, PPV4 or PPV5 was detected in 3.8% of samples (3/80). With regard to multiple infection, PPV2+PPV5 showed the highest rate (6.3%, 5/80). Not taking into account multiple infection, the simple infection rate was highest for PPV6 (30.0%, 24/80), followed by PPV2 (21.3%, 17/80), PPV5 (18.8%, 15/80), PPV4 (16.3%, 13/80) and PPV3 (5.0%, 4/80). PPV1 and PPV7 were not detected in serum samples (Fig 4).

Among lung/lymph node samples, 67.5% (27/40) of the samples were virus positive. PPV3 and PPV7 showed the highest single infection rates (7.5%, 3/40). Interestingly, the multiple infection rate (47.5%, 19/40) was higher than the single infection rate (20.0%, 8/40) in lung and lymph node samples. PPV6 showed the highest simple infection rate (27.5%, 11/40), followed by PPV2 and PPV3 (25.0%, 10/40); PPV1, PPV5 and PPV7 (20.0%, 8/40); and PPV4 (7.5%, 3/40) (Fig 4).

Among intestine and fecal samples, 16 of the 40 samples were virus positive (40%). PPV3 exhibited the highest single infection rate (10.0%, 4/40), followed by PPV2 (7.5%, 3/40). The multiple infection rate (17.5%, 7/40) was slightly lower than the single infection rate (22.5%, 9/40) in intestine/fecal samples. The simple infection rates were 17.5% (7/40) for PPV3, 15.0% (6/40) for PPV2, 12.5% (5/40) for PPV7, 10.0% (4/40) for PPV6, and 2.5% (1/40) for both PPV4 and PPV5. PPV1 was not detected (Fig 4).

**Table 2. Detection of PPV types from various types of field samples from Korea using the PPV mPCR method.**

| | Serum (n = 80) | | Lung & LN (n = 40) | | IN & Feces (n = 40) | |
|---|---|---|---|---|---|---|
| | $n_{pos}$ (a) | rate(b) | $n_{pos}$ | rate | $n_{pos}$ | rate |
| **Single infection** | | | | | | |
| PPV1 | 0 | 0.00% | 0 | 0.00% | 0 | 0.00% |
| PPV2 | 9 | 11.30% | 1 | 2.50% | 3 | 7.50% |
| PPV3 | 3 | 3.80% | 3 | 7.50% | 4 | 10.00% |
| PPV4 | 3 | 3.80% | 0 | 0.00% | 1 | 2.50% |
| PPV5 | 3 | 3.80% | 0 | 0.00% | 0 | 0.00% |
| PPV6 | 10 | 12.50% | 1 | 2.50% | 1 | 2.50% |
| PPV7 | 0 | 0.00% | 3 | 7.50% | 0 | 0.00% |
| **Total** | 28 | 35.00% | 8 | 20.00% | 9 | 22.50% |
| **Multiple infections** | | | | | | |
| PPV2+PPV3 | 0 | 0.00% | 1 | 2.50% | 0 | 0.00% |
| PPV2+PPV4 | 0 | 0.00% | 2 | 5.00% | 0 | 0.00% |
| PPV2+PPV5 | 5 | 6.30% | 1 | 2.50% | 0 | 0.00% |
| PPV2+PPV6 | 2 | 2.50% | 1 | 2.50% | 0 | 0.00% |
| PPV2+PPV7 | 0 | 0.00% | 2 | 5.00% | 2 | 5.00% |
| PPV3+PPV5 | 0 | 0.00% | 0 | 0.00% | 1 | 2.50% |
| PPV3+PPV6 | 0 | 0.00% | 1 | 2.50% | 1 | 2.50% |
| PPV4+PPV5 | 1 | 1.30% | 0 | 0.00% | 0 | 0.00% |
| PPV4+PPV6 | 4 | 5.00% | 0 | 0.00% | 0 | 0.00% |
| PPV5+PPV6 | 3 | 3.80% | 1 | 2.50% | 0 | 0.00% |
| PPV5+PPV7 | 0 | 0.00% | 1 | 2.50% | 0 | 0.00% |
| PPV6+PPV7 | 0 | 0.00% | 0 | 0.00% | 2 | 5.00% |
| PPV1+PPV2+PPV3 | 0 | 0.00% | 1 | 2.50% | 0 | 0.00% |
| PPV1+PPV3+PPV6 | 0 | 0.00% | 2 | 5.00% | 0 | 0.00% |
| PPV1+PPV5+PPV6 | 0 | 0.00% | 2 | 5.00% | 0 | 0.00% |
| PPV2+PPV3+PPV7 | 0 | 0.00% | 0 | 0.00% | 1 | 2.50% |
| PPV2+PPV4+PPV6 | 1 | 1.30% | 0 | 0.00% | 0 | 0.00% |
| PPV3+PPV4+PPV6 | 1 | 1.30% | 0 | 0.00% | 0 | 0.00% |
| PPV4+PPV5+PPV6 | 3 | 3.80% | 0 | 0.00% | 0 | 0.00% |
| PPV1+PPV3+PPV5+PPV6 | 0 | 0.00% | 1 | 2.50% | 0 | 0.00% |
| PPV1+PPV2+PPV4+PPV5+PPV6+PPV7 | 0 | 0.00% | 1 | 2.50% | 0 | 0.00% |
| **Total** | 20 | 25.00% | 19 | 47.50% | 7 | 17.50% |

a$n_{pos}$, number of positive samples

brate, positive rate

## Discussion

The roles of novel PPVs in swine diseases are not completely understood. In addition, only PPV1 has been cultured in vitro, and virus challenge experiments for the other novel PPVs have not been performed. Therefore, the significance of PPVs can be inferred only at the DNA level [4]. Previous studies have investigated the prevalence of PPVs but have not simultaneously tested all the PPV types, PPV1 through PPV7 [4, 11, 16], as such investigation would cost considerable time and effort. In addition, no previous study has developed a diagnostic method to simultaneously detect PPV1 through PPV7. Here, we developed an mPCR method

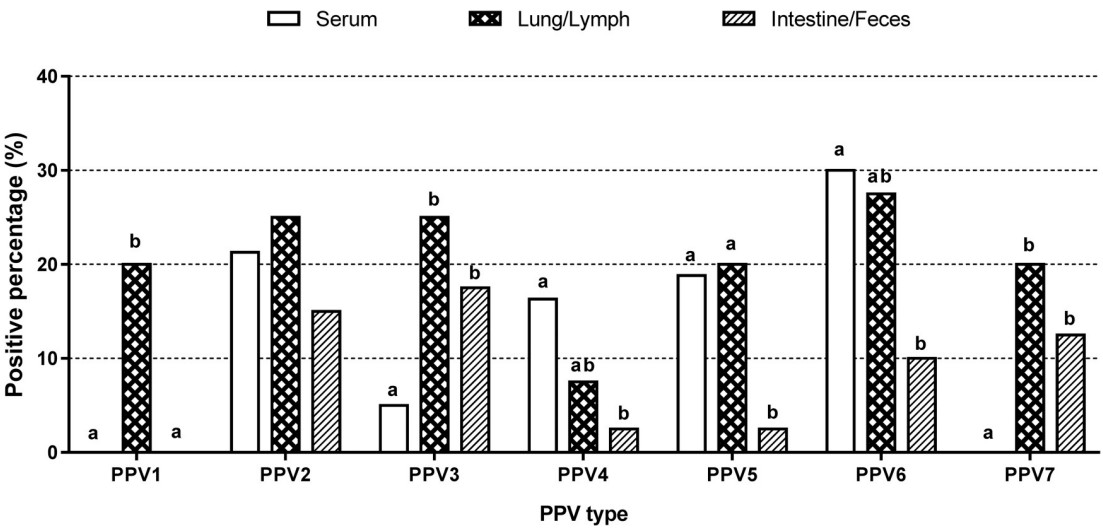

**Fig 4. Simple infection rates of each PPV type within different types of samples.** Percentages of PPV1- to PPV7-positive serum samples (n = 80), lung/lymph node samples (n = 40), and intestine/fecal samples (n = 40). Statistically significant differences ($p < 0.05$, Fisher's exact test) between types of samples within the grouped bars for each virus are marked with superscripts on the top of bars in the chart (a, b).

for the concurrent detection of PPVs. To our knowledge, this method is the first developed method that is capable of detecting PPV1 through PPV7.

Primer design is the first and critical step in the process of establishing an mPCR assay [28]. Primers that are designed manually should satisfy 3 criteria: (1) they should target specific and highly conserved regions for each PPV type, (2) they should have similar annealing temperatures, and (3) they should not self-dimerize or create cross-primer dimers and thus should not generate nonspecific bands [20]. In the current study, primers were designed using complete sequences from GenBank to target the seven types of PPVs based on a conserved region. The primer sets for PPV1, PPV2, PPV6 and PPV7 targeted the NS1 genes, while the primer sets for PPV3 and PPV5 targeted the VP genes. The primers for PPV4 were designed to target the ORF3 region because only PPV4 has a characteristic ORF3 region (Table 1). The combinations of primers produced amplicons that could easily be used to distinguish among the different types of PPVs. Moreover, the primer annealing temperatures (Tas) were similar, and there were no self-dimers, cross-primer dimers or nonspecific bands.

The establishment of a new mPCR assay is a complicated task because the existence of more than one primer pair in the same reaction mix may limit the sensitivity or specificity [29, 30]. However, the specific primers in this study produced distinct amplicons for each PPV type, which could be visualized and easily differentiated by 2% agarose gel electrophoresis. A cross-reactivity test for pathogens that could be present due to simultaneous infection or secondary infection, including viruses (PRRSV, PCV2, PED, TGE, rotavirus, JEV, PRV and CSFV) and bacteria (*E. coli*, *Salmonella enterica*), showed that the proposed new mPCR assay had high specificity (Fig 2). In a previous study, the detection limit of a single conventional PCR method for PPV4 was approximately $9.5 \times 10^2$ copies [22]. In addition, an mPCR method for detecting porcine bocavirus (PBoV), a virus similar to PPVs, shows detection limits of $1.0 \times 10^3$, $4.5 \times 10^3$, and $3.8 \times 10^3$ copies/µl for PBoV G1, G2, and G3, respectively [31]. Likewise, the new mPCR assay developed in the current study was able to detect as few as $3 \times 10^3$ copies/µl of the seven types of PPVs, indicating that it showed high sensitivity (Fig 3).

A total of 160 samples (serum: 80, lung/lymph node: 40, intestine/feces: 40) from JBNU-VDC were tested using the new mPCR method. The results showed that PPVs were readily detected regardless of the sample type and that single PCR amplicons were easily distinguished for each type of PPV. Twenty-eight PPV-positive samples were randomly selected for comparison of the mPCR method with single PCR methods. The results indicated that the positivity rate of the new mPCR assay and the single PCR methods exhibited 100% agreement, although the mPCR method showed slightly lower positivity rates for PPV2 and PPV3 and slightly higher positivity rates for PPV6 and PPV7 than the single PCR methods. Therefore, the newly developed mPCR assay enables faster detection of various PPV types than other established PCR methods.

In previous studies, although the detection rates of PPVs have varied among reports by sample type and age, all types of PPVs have been able to be identified from serum, lung, oral fluid and fecal samples [4, 11, 16, 32]. Despite the limited total sample size in this study, all types of PPVs were successfully detected by using the new mPCR method in serum, lung/lymph node and intestine/fecal samples, except for PPV1 in serum and intestine/fecal samples and PPV7 in serum samples (Fig 4). This result indicates that the mPCR method established in this study is effective for detection of PPV DNA in various sample types; thus, the main aim of this study was accomplished. The prevalence of each PPV by age and sample type should be investigated and analyzed in detail in future studies.

mPCR is designed to detect more than one target and has the potential to considerably reduce the time and effort necessary for detection. Moreover, mPCR has been successfully used for molecular diagnostics at various diagnostic laboratories [33–36]. Although fluorogenic real-time PCR assays have many advantages over conventional PCR and mPCR assays at diagnostic laboratories in developed countries, fluorogenic PCR is still being developed, and its cost prohibits its use in many developing countries [33]. Additionally, probe-based single or multiplex TaqMan real-time PCR assays are costly and highly influenced by the potential existence of mutations in the probe binding site that can prevent probe annealing and subsequent detection [37]. However, the new mPCR method developed in the present study successfully detected the various PPV types and has the potential to save time, effort and reagent costs for routine diagnosis of PPVs. Furthermore, it reduces the amount of sample required for the assay, which is beneficial when sample material is limited. To our knowledge, this is the first report of the development of a simple, sensitive and specific assay to detect and differentiate PPV1 through PPV7 in various sample types for monitoring and diagnosis of PPV infections in swine. Therefore, the developed mPCR method can be used as a valuable tool with which to investigate the prevalence or circulation patterns of novel PPVs in the field, as it requires less effort and exhibits higher efficiency than existing methods.

## Conclusion

In conclusion, this paper describes the development and evaluation of an mPCR method that has been demonstrated to enable rapid, sensitive, specific and concurrent detection of the different PPV genotypes in various types of samples. The newly developed mPCR assay can simplify the diagnostic procedure and reduce reagent and labor costs.

## Supporting information

**S1 Fig. Reproducibility of the PPV mPCR method.** Mixed PPV plasmids were diluted as templates from $3\times10^5$ to $3\times10^3$ copies/μl to amplify specific fragments by using three different PCR instruments at different times. Lane M, 100-bp plus DNA ladder; lanes 1, 5, and 9, negative controls for the three tests; lanes 2~4, mPCR amplification of $3\times10^5$ copies/μl, $3\times10^4$

copies/μl, and 3×10$^3$ copies/μl PPV plasmids; lanes 6~8 and 10~12, repeated mPCR.
(PDF)

**S2 Fig. Evaluation of field samples.** Eighty serum, 40 lung/lymph node and 40 intestine/fecal samples were tested with the established mPCR method. (A) Serum samples: lane M, 100 bp-plus DNA ladder; lane PC, positive controls; lane NC, negative controls; lanes 1~80, mPCR-tested serum samples. (B) Lung/lymph node samples: lane M, 100 bp-plus DNA ladder; lane PC, positive controls; lane NC, negative controls; lanes 1~40, mPCR-tested lung/lymph node samples. (C) Intestine/fecal samples: lane M, 100 bp-plus DNA ladder; lane PC, positive controls; lane NC, negative controls; lanes 1~40, mPCR-tested intestine/fecal samples.
(PDF)

**S1 Table. NCBI GenBank sequences used in this study for primer design.**
(XLSX)

**S2 Table. Reference primers used for validation of the PPV mPCR method in this study.**
(XLSX)

## Acknowledgments

The authors wish to acknowledge the assistance of the laboratory technicians from the Jeonbuk National University Veterinary Diagnostic Center (JBNU-VDC).

## Author Contributions

**Conceptualization:** Seung-Chai Kim, Salik Nazki.

**Data curation:** Seung-Chai Kim, Salik Nazki.

**Formal analysis:** Seung-Chai Kim.

**Funding acquisition:** Won-Il Kim.

**Investigation:** Seung-Chai Kim, Sim-In Lee, Ye-Chan Baek, Yong-Jin Jung.

**Methodology:** Seung-Chai Kim, Salik Nazki.

**Resources:** Sim-In Lee, Ye-Chan Baek, Yong-Jin Jung.

**Supervision:** Won-Il Kim.

**Validation:** Chang-Gi Jeong.

**Writing – original draft:** Seung-Chai Kim, Chang-Gi Jeong.

**Writing – review & editing:** Won-Il Kim.

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
