## [Decision Letter · Decision Letter 0]

18 Dec 2020

PONE-D-20-32226

Evaluation of a multiplex PCR method for the detection of porcine parvovirus types 1 through 7 with various field samples

PLOS ONE

Dear Dr. Kim

Thank you for submitting your manuscript to PLOS ONE. After careful consideration, we feel that it has merit but does not fully meet PLOS ONE’s publication criteria as it currently stands. Therefore, we invite you to submit a revised version of the manuscript that addresses the points raised during the review process.

Many thanks for submitting your manuscript to PLOS One

Your manuscript was reviewed by three experts in the field who have recommended some modifications be made prior to acceptance

I therefore invite you to make these changes, and write a brief response to reviewers to aid review when you re-submit

I wish you the best of luck with your changes

Hope you are keeping safe and well in these difficult times

We look forward to receiving your revised manuscript.

Kind regards,

Simon Clegg, PhD

Academic Editor

PLOS ONE

Reviewers' comments:

Reviewer's Responses to Questions

**Comments to the Author**

1. Is the manuscript technically sound, and do the data support the conclusions?

Reviewer #1: Yes

Reviewer #2: Yes

Reviewer #3: Yes

2. Has the statistical analysis been performed appropriately and rigorously? 

Reviewer #1: Yes

Reviewer #2: Yes

Reviewer #3: N/A

3. Have the authors made all data underlying the findings in their manuscript fully available?

Reviewer #1: No

Reviewer #2: Yes

Reviewer #3: Yes

4. Is the manuscript presented in an intelligible fashion and written in standard English?

Reviewer #1: No

Reviewer #2: Yes

Reviewer #3: No

5. Review Comments to the Author

Reviewer #1: The diagnosis method for PPV 1 to 7 is very important and useful. And this research was focus on multiple PCR for PPV 1 to 7 at the same time. And the Specificity, sensitivity, and reproducibility of multiple PCR method was OK. But there were some faults in this manuscript.

Main suggestion:

1. The PPV genome sequenced by the author should be shown in the manuscript.

2. PRV and CSFV maybe should be added as negative control.

3. Lack of compare with golden standard, such as virus isolation and genome sequence of PPV 1 to PPV7 from clinical samples.

4. Figure 4 should be moved to supplemental as Figure S1.

5. Table 2 should be moved to supplemental as Table S2.

6. Table S2 should be moved to main manuscript as Table 2.

7. The sample number is too low (80 serum samples, 40 lung or lymph node samples and 40 intestine or fecal samples), perhaps more samples should be detected.

Reviewer #2: This was a generally well-written paper and quite enjoyable to read! The development of a multiplex PCR assay for the diagnosis of porcine parvovirus types 1-7 certainly solves an important problem with it's rapidity and synchronicity. There are a few minor concerns that I'd like some clarification on or have some suggestions for.

The abstract and introduction read well and largely have the relevant information, but I would perhaps recommend including the clinical signs associated with PPV in the latter?

The methods seems quite repeatable with a good level of detail, but I'd just like to clarify:

1. What are the origins of the virus isolates that were grown (PRRSV-1 and PCV2 on page 5)?

2. Would the sample information (pages 5-6) be better presented in a table with what kind of animals (foetuses, piglets, etc) they came from and the clinical signs they may have had?

3. What is the approximate level of circulation or prevalence of the 7 different strains? Is PPV4 less common, and thus has less sequences that were used, than PPV7 (page 6)?

4. Why were a variety of genes targeted in Table 1 rather than just one, such as the VP gene in Table 2?

5. I'm intrigued why 60 samples in total (with no intestine or faecal samples) were used, were these known to be positive (page 7) and were negative controls used? Were other techniques (viral isolation, IPMA, IF assay, etc) performed to confirm positives or compare results?

The results were again quite coherent, I would just like to ask:

1. Could the temperatures and labels (DNA ladder, PK-15, PGEM-PPV1, etc) be put in place of the numbering system (M, 1, 2, 3...) on the figures, which will then shorten the captions and be easier to visualise?

2. Would the cross-validation results (page 13) be better as a table with percentages, again for quick visualisation, especially of how the mPCR assay is superior than simplex PCR for PPV6 and 7?

The discussion and conclusion were again well-written and summarised. Overall, a very beneficial diagnostic development!

Reviewer #3: Reviewer’s report

Evaluation of a multiplex PCR method for the detection of porcine parvovirus types 1 through 7 with various field samples by Kim et al.

This research article of Kim et al. described a multiplex PCR method for the simultaneous detection of porcine parvoviruses type 1 through 7 using different field samples. The authors were able to prove beyond reasonable doubt, the efficacy of their multiplex PCR method in detecting and differentiating various known PPVs. With the use of standard methods, the authors were able to do justice to the aim of the study and produced substantial amount of data that corroborate their findings. The development of the multiplex PCR method for simultaneous detection of PPVs is commendable, there is no doubt that it will aid a faster and cost-effective surveillance of the swine viruses. Equally, the research work is relatively novel as many of the published multiplex PCR methods focused on detecting and differentiating different swine viruses and not PPVs as described in this manuscript. Generally, the research work was well executed and the manuscript was well written, except that some errors need to be corrected and some ambiguous statements requires clarification/revision.

Reviewer's comments and observations:

Title: The authors should consider changing the world “with” to “using”

INTRODUCTION SECTION

Paragraph 1, Line 4, page 3: The Parvovirinae family comprises “eight” genera as stated, not nine.

Paragraph 3, Line 1/2, page 3: The statement: “PPVs are ubiquitous in the global pig population and are major causative agents of reproductive failure in pigs.” is misleading. Out of all the PPVs, only PPV1 has been well established as the causative agent of reproductive failure in pigs, not all the PPVs. It will be good for the authors to be specific and should not make outrageous general statements.

Paragraph 3, Line 2/3, page 3/4: Same as above, the statement: “Moreover, these viruses are known to contribute to the development of porcine circovirus-associated disease (PCVAD).” seems to be too assertive. Although PPV1 co-infection with PCV2 has been implicated in the development of PCVAD, the involvement of other novel PPVs in the disease has not been well established. Hence, the author should specifically mention the disease conditions in which each of the PPVs have been implicated, and should not make incorrect general statement.

Paragraph 3, Line 6, page 4: The statement: “…simultaneous detection of various pathogens…” should be written as “…simultaneous detection of various swine pathogens…”

MATERIALS AND METHODS SECTION

Sub-section: Cells, viruses and bacterial strains

Third to the last line: The manufacturer of Luria-Bertani (LB) agar and LB broth should be stated.

Sub-section: Samples and nucleic acid preparation

Paragraph 2, Line 1, page 6: “Viral nucleic acids were extracted from the serum samples using…” How? According to the manufacturer's instruction or how? This should be well stated. How many mL of the serum samples was used?

Paragraph 2, Line 3, page 6: “…were homogenized…” Homogenized, how? How many grams?

Sub-section: Optimization of the mPCR method

Line 3, page 8: “The reaction for detection of all…” should be revised as “The reaction for detecting all…”

RESULTS SECTION

Sub-section: Evaluation of field samples

In general, I would like to suggest that the table S2 should not be supplementary, it should be presented as substantive table in the manuscript. References were made to the contents of the table severally under this sub-section, hence, it should not be supplementary.

Paragraph 1, Line 2, page 13: Article “The” should be placed before “mPCR”

Paragraph 2, Line 3, page 14: The word “and” should be replaced with “followed by”

Paragraph 3, Line 2/3, page 14: There is a serious mix-up in the sentence: “Interestingly, the multiple infection rate (20.0%, 8/40) was higher than the single infection rate (47.5%, 19/40) in lung and lymph node samples.” The mix-up should be checked and corrected from the table.

DISCUSSION SECTION

Paragraph 1, Line 1, page 16: The statement: “…swine health disorders…” should be corrected as “…swine diseases…”

Paragraph 1, Line 2/3, page 16: The statement: “…and virus challenge experiments have not been able to be performed.” should be properly revised to make better meaning.

Paragraph 2, Line 11, page 16: The joined “temperaturesTas” should be separated and the “Tas” should be put in bracket.

Paragraph 5, Line 1, page 17: In the statement: “although the detection rates of PPVs have varied by sample type and age…” Have varied what? It seems an information is missing.

Paragraph 6, Line 11/12, page 18: The statement: “Furthermore, it reduces the sample amount required for the assay…” should be revised as “Furthermore, it reduces the amount of sample required for the assay…”

Paragraph 6, Line 16, page 18: The statement: “the prevalence or circulation patterns of novel PPVs in the field that requires less effort…” should be revised as “the prevalence or circulation patterns of novel PPVs in the field, as it requires less effort…”

CONCLUSION SECTION

Line 3, page 19: The statement: “different PPV types in various types of samples…” should be revised as “different PPV genotypes in various types of samples…”

ACKNOWLEDGMENTS SECTION

The information: “…and undergraduate students from the College of Veterinary Medicine, Veterinary Diagnostic” seems to be incomplete.

6. PLOS authors have the option to publish the peer review history of their article (what does this mean?). If published, this will include your full peer review and any attached files.

Reviewer #1: No

Reviewer #2: No

Reviewer #3: No

---

## [Author Response · Author response to Decision Letter 0]

5 Jan 2021

Response to Reviewers

Reviewer #1: The diagnosis method for PPV 1 to 7 is very important and useful. And this research was focus on multiple PCR for PPV 1 to 7 at the same time. And the Specificity, sensitivity, and reproducibility of multiple PCR method was OK. But there were some faults in this manuscript.

Main suggestion:

1. The PPV genome sequenced by the author should be shown in the manuscript.

Response: The sequences of each PPV genotype PCR amplicons, which were cloned into the pGEM-T Easy vector and used as positive controls, were submitted to the NCBI GenBank under the accession number MW401540-MW401546. Corresponding information was added to the manuscript (“Construction of plasmids for positive controls” section, paragraph 2, lines 11-12).

2. PRV and CSFV maybe should be added as negative control.

Response: As suggested, PRV and CSFV were added as negative control, and the mPCR assay was performed again. Corresponding contents were added into the Materials & Methods and Results section based on (revised) Figure 2.

3. Lack of compare with golden standard, such as virus isolation and genome sequence of PPV 1 to PPV7 from clinical samples.

Response: Novel PPVs (PPV2~PPV7) were identified recently and have never been isolated in cell culture (Milek et al., 2019, which was also added to the Introduction section). In addition, no commercial antibodies were available for novel PPVs. Thus, the only way to confirm PPV1~PPV7 was based on the size of amplified bands and sequencing of those bands for confirmation through the BLAST search. 

4. Figure 4 should be moved to supplemental as Figure S1.

Response: Figure 4 has been moved to the supplemental information as Figure S1, as suggested. 

5. Table 2 should be moved to supplemental as Table S2.

Response: Table 2 has been moved to the supplemental information as Table S2, as suggested. 

6. Table S2 should be moved to main manuscript as Table 2.

Response: Table S2 has been moved to the supplemental information as Table 2, as suggested. 

7. The sample number is too low (80 serum samples, 40 lung or lymph node samples and 40 intestine or fecal samples), perhaps more samples should be detected.

Response: The main goal of this study was to establish the multiplex PCR (mPCR) method for PPV1~PPV7, which can be applied to various sample types (serum, lungs, feces, etc.). The authors prioritized rapidity and applicability for different sample type. Novel PPVs (PPV2~PPV7) were found from both unhealthy and healthy pigs and from various age stage, which make performing an epidemiological study of PPVs difficult. In addition, the sample size of this study is not adequate for an epidemiological study, which should be performed cautiously. However, the sample number was sufficient for evaluation using the new mPCR technique. The established mPCR method from this study will be applied for the epidemiological investigation of PPVs in Korea in a future study. The prevalence study is in progress.

Reviewer #2: This was a generally well-written paper and quite enjoyable to read! The development of a multiplex PCR assay for the diagnosis of porcine parvovirus types 1-7 certainly solves an important problem with its rapidity and synchronicity. There are a few minor concerns that I'd like some clarification on or have some suggestions for.

The abstract and introduction read well and largely have the relevant information, but I would perhaps recommend including the clinical signs associated with PPV in the latter?

Response: The clinical signs of PPV1 were described in the Introduction as suggested (lines 57~63, page 3~4)

The methods seems quite repeatable with a good level of detail, but I'd just like to clarify:

1. What are the origins of the virus isolates that were grown (PRRSV-1 and PCV2 on page 5)?

Response: PRRSV1 isolate CBNU0495 (KY434183.1) is a Korean field PRRSV1 strain, and PCV2 isolate is also a Korean field isolate. These isolates were isolated and sustained in our lab. Field samples containing the isolates mentioned above were subjected to Jeonbuk National University Veterinary Diagnostic Center (JBNU-VDC) in the past, and the viruses were isolated. 

2. Would the sample information (pages 5-6) be better presented in a table with what kind of animals (foetuses, piglets, etc) they came from and the clinical signs they may have had?

Age group Sample type Total

 Serum Lung/

lymph node Intestine/

feces 

Suckling 8 0 7 15

Weaned 22 19 19 60

Growing 13 11 9 33

Finisher 9 10 3 22

Gilt/Sow 28 0 2 30

Total 80 40 40 

Response: The main goal of this study was to establish the multiplex PCR (mPCR) method for PPV1~PPV7, which can be applied to various sample types (serum, lungs, feces, etc.). The authors prioritized rapidity and applicability for different sample types. The age-wise difference or health condition will be further investigated and studied. Also, a small number of samples was used in this study, and most of the samples were obtained from weaned or growing piglets. 

Novel PPVs (PPV2~PPV7) were identified from both unhealthy and healthy pigs and from various age stage, which makes performing an epidemiological study of PPVs difficult. Plus, the sample size of this study is not adequate for an epidemiological study, which must be performed with caution. However, the sample size is sufficient for evaluation using this new mPCR method. The established mPCR method from this study will be applied for epidemiological investigations of PPVs in Korea. The prevalence study is in progress.

3. What is the approximate level of circulation or prevalence of the 7 different strains? Is PPV4 less common, and thus has less sequences that were used, than PPV7 (page 6)?

Response: Given that novel PPVs (PPV2~PPV6) were recently identified, the number of complete genome sequences of each PPVs archived in NCBI GenBank varied based on the time this study was conducted. In addition, the description of complete sequences of each PPVs, which were submitted to GenBank in early years after their discovery, were vague (especially for PPV3, which was also called porcine PARV4, hokovirus and partetravirus). Some groups even submitted the complete genome of Porcine PARV4 (PPV3) as PPV4 to GenBank. As a result, complete genome sequences were selected only when their sequences were trustworthy, and the number of total sequences used to design primers for each PPV types differed.

4. Why were a variety of genes targeted in Table 1 rather than just one, such as the VP gene in Table 2?

Response: The nonstructural protein gene region (NS) of PPVs are highly conserved for each genera, whereas the viral capsid gene region is more diverse and acts as a genetic determinants of virulence. Thus, the primers were initially designed by targeting the NS region for each PPVs. However, as the goal was to design multiplex primers, designing primers which amplifies not only specific NS region of each PPVs but also different 100~150-bp bands for each type was not easy. First, the NS region of PPV4, PPV5 and PPV6 were close to each other as they are classified into Copiparvovirus genus altogether. Plus, the primer designed to target the PPV3 NS region amplified an unwanted band from PPV4. As a result, primers for PPV3, PPV4, PPV5 and PPV6 were re-deigned with caution. For PPV3 and PPV5, the conserved region in VP was investigated and used for primer design. PPV4 has a unique additional ORF3 that differs from other parvoviruses and was utilized for primer. Luckily, PPV6 contained a unique conserved region that differed from the NS region of other Copiparvovirus genera (PPV4 and PPV5), and the primer was designed. In conclusion, by targeting different genes (but conserved), all primers could be successfully designed to yield an adequate band size to distinguish seven viruses at once. 

5. I'm intrigued why 60 samples in total (with no intestine or faecal samples) were used, were these known to be positive (page 7) and were negative controls used? Were other techniques (viral isolation, IPMA, IF assay, etc) performed to confirm positives or compare results?

Response: Novel PPVs (PPV2~PPV7) were identified recently and have never been isolated in cell culture (Milek et al., 2019, which was also added to the Introduction section). In addition, no commercial antibodies were available for novel PPVs. However, numerous studies reported that novel PPVs are already prevalent in the global pig population. Consequently, 60 field samples mentioned on page 7 were first subject to screening by simplex PCR using newly designed primers. Among these samples, amplified bands were cloned into the pGEM-T Easy vector and sent for sequencing to confirm whether it is a part of targeted PPV genome. In conclusion, the only method to confirm positive findings is sequencing.

The results were again quite coherent, I would just like to ask:

1. Could the temperatures and labels (DNA ladder, PK-15, PGEM-PPV1, etc) be put in place of the numbering system (M, 1, 2, 3...) on the figures, which will then shorten the captions and be easier to visualise?

Response: We tried to do this, but the names (e.g., pGEM-PPV1) were too long to add to the position on top of each well of the gel picture. Thus, the numbering system was the best option.

2. Would the cross-validation results (page 13) be better as a table with percentages, again for quick visualisation, especially of how the mPCR assay is superior than simplex PCR for PPV6 and 7?

Response: The authors modified the cross-validation results to demonstrate 100% agreement between mPCR and simplex PCRs for the total number of tests (lines 253~255, page 13). Given that a small number of samples were used for each PPV type, the authors were concerned that presentation in a table could be exaggerated.

The discussion and conclusion were again well-written and summarised. Overall, a very beneficial diagnostic development!

Reviewer #3: Reviewer’s report

Evaluation of a multiplex PCR method for the detection of porcine parvovirus types 1 through 7 with various field samples by Kim et al.

This research article of Kim et al. described a multiplex PCR method for the simultaneous detection of porcine parvoviruses type 1 through 7 using different field samples. The authors were able to prove beyond reasonable doubt, the efficacy of their multiplex PCR method in detecting and differentiating various known PPVs. With the use of standard methods, the authors were able to do justice to the aim of the study and produced substantial amount of data that corroborate their findings. The development of the multiplex PCR method for simultaneous detection of PPVs is commendable, there is no doubt that it will aid a faster and cost-effective surveillance of the swine viruses. Equally, the research work is relatively novel as many of the published multiplex PCR methods focused on detecting and differentiating different swine viruses and not PPVs as described in this manuscript. Generally, the research work was well executed and the manuscript was well written, except that some errors need to be corrected and some ambiguous statements requires clarification/revision.

Reviewer's comments and observations:

Title: The authors should consider changing the world “with” to “using”

Response: It was corrected as suggested.

INTRODUCTION SECTION

Paragraph 1, Line 4, page 3: The Parvovirinae family comprises “eight” genera as stated, not nine.

Response: It was corrected as suggested.

Paragraph 3, Line 1/2, page 3: The statement: “PPVs are ubiquitous in the global pig population and are major causative agents of reproductive failure in pigs.” is misleading. Out of all the PPVs, only PPV1 has been well established as the causative agent of reproductive failure in pigs, not all the PPVs. It will be good for the authors to be specific and should not make outrageous general statements.

Response: The sentence has been revised as follows: “PPV1 was the sole representative of Parvovirinae members until recently and has been ubiquitous in the global pig population as a major causative agent of reproductive failure in pigs.”

Paragraph 3, Line 2/3, page 3/4: Same as above, the statement: “Moreover, these viruses are known to contribute to the development of porcine circovirus-associated disease (PCVAD).” seems to be too assertive. Although PPV1 co-infection with PCV2 has been implicated in the development of PCVAD, the involvement of other novel PPVs in the disease has not been well established. Hence, the author should specifically mention the disease conditions in which each of the PPVs have been implicated, and should not make incorrect general statement.

Response: The statement has been removed.

Paragraph 3, Line 6, page 4: The statement: “…simultaneous detection of various pathogens…” should be written as “…simultaneous detection of various swine pathogens…”

Response: It was corrected as suggested.

MATERIALS AND METHODS SECTION

Sub-section: Cells, viruses and bacterial strains

Third to the last line: The manufacturer of Luria-Bertani (LB) agar and LB broth should be stated.

Response: It was corrected as suggested.

Sub-section: Samples and nucleic acid preparation

Paragraph 2, Line 1, page 6: “Viral nucleic acids were extracted from the serum samples using…” How? According to the manufacturer's instruction or how? This should be well stated. How many mL of the serum samples was used?

Response: It was corrected as suggested.

Paragraph 2, Line 3, page 6: “…were homogenized…” Homogenized, how? How many grams?

Response: It was corrected as suggested.

Sub-section: Optimization of the mPCR method

Line 3, page 8: “The reaction for detection of all…” should be revised as “The reaction for detecting all…”

Response: It was corrected as suggested.

RESULTS SECTION

Sub-section: Evaluation of field samples

In general, I would like to suggest that the table S2 should not be supplementary, it should be presented as substantive table in the manuscript. References were made to the contents of the table severally under this sub-section, hence, it should not be supplementary.

Response: Table S2 has been moved to Table 2, as suggested.

Paragraph 1, Line 2, page 13: Article “The” should be placed before “mPCR”

Response: It was corrected as suggested.

Paragraph 2, Line 3, page 14: The word “and” should be replaced with “followed by”

Response: It was corrected as suggested.

Paragraph 3, Line 2/3, page 14: There is a serious mix-up in the sentence: “Interestingly, the multiple infection rate (20.0%, 8/40) was higher than the single infection rate (47.5%, 19/40) in lung and lymph node samples.” The mix-up should be checked and corrected from the table.

Response: It was corrected as suggested.

DISCUSSION SECTION

Paragraph 1, Line 1, page 16: The statement: “…swine health disorders…” should be corrected as “…swine diseases…”

Response: It was corrected as suggested.

Paragraph 1, Line 2/3, page 16: The statement: “…and virus challenge experiments have not been able to be performed.” should be properly revised to make better meaning.

Response: The manuscript has been fixed as suggested by adding the comment “…and virus challenge experiments for the other novel PPVs have not been performed.”

Paragraph 2, Line 11, page 16: The joined “temperaturesTas” should be separated and the “Tas” should be put in bracket.

Response: It was corrected as suggested.

Paragraph 5, Line 1, page 17: In the statement: “although the detection rates of PPVs have varied by sample type and age…” Have varied what? It seems an information is missing.

Response: It was corrected, as suggested by adding the comment “although the detection rates of PPVs have varied among reports by sample type and age…”

Paragraph 6, Line 11/12, page 18: The statement: “Furthermore, it reduces the sample amount required for the assay…” should be revised as “Furthermore, it reduces the amount of sample required for the assay…”

Response: It was corrected as suggested.

Paragraph 6, Line 16, page 18: The statement: “the prevalence or circulation patterns of novel PPVs in the field that requires less effort…” should be revised as “the prevalence or circulation patterns of novel PPVs in the field, as it requires less effort…”

Response: It was corrected as suggested.

CONCLUSION SECTION

Line 3, page 19: The statement: “different PPV types in various types of samples…” should be revised as “different PPV genotypes in various types of samples…”

Response: It was corrected as suggested.

ACKNOWLEDGMENTS SECTION

The information: “…and undergraduate students from the College of Veterinary Medicine, Veterinary Diagnostic” seems to be incomplete.

Response: It was corrected as suggested.

---

## [Editor Report · Decision Letter 1]

6 Jan 2021

Evaluation of a multiplex PCR method for the detection of porcine parvovirus types 1 through 7 using various field samples

PONE-D-20-32226R1

Dear Dr. Kim,

We’re pleased to inform you that your manuscript has been judged scientifically suitable for publication and will be formally accepted for publication once it meets all outstanding technical requirements.

Kind regards,

Simon Clegg, PhD

Academic Editor

PLOS ONE

Additional Editor Comments:

Many thanks for resubmitting your manuscript to PLOS One

As you have addressed all the comments and the manuscript reads well, I have recommended it for publication

You should hear from the Editorial Office shortly.

It was a pleasure working with you and I wish you the best of luck for your future research

Hope you are keeping safe and well in these difficult times

Thanks

Simon

---

## [Editor Report · Acceptance letter]

8 Jan 2021

PONE-D-20-32226R1 

Evaluation of a multiplex PCR method for the detection of porcine parvovirus types 1 through 7 using various field samples 

Dear Dr. Kim:

I'm pleased to inform you that your manuscript has been deemed suitable for publication in PLOS ONE. Congratulations! Your manuscript is now with our production department. 

Kind regards, 

on behalf of

Dr. Simon Clegg 

Academic Editor

PLOS ONE